# Working Memory Performance under a Negative Affect Is More Susceptible to Higher Cognitive Workloads with Different Neural Haemodynamic Correlates

**DOI:** 10.3390/brainsci11070935

**Published:** 2021-07-15

**Authors:** Ying Xing Feng, Masashi Kiguchi, Wei Chun Ung, Sarat Chandra Dass, Ahmad Fadzil Mohd Hani, Tong Boon Tang, Eric Tatt Wei Ho

**Affiliations:** 1Centre for Intelligence Signal and Imaging Research (CISIR), Universiti Teknologi PETRONAS, Bandar Seri Iskandar 32610, Perak, Malaysia; yx.feng@outlook.com (Y.X.F.); ungweichun@gmail.com (W.C.U.); tongboon.tang@utp.edu.my (T.B.T.); 2Department of Electrical & Electronics Engineering, Universiti Teknologi PETRONAS, Bandar Seri Iskandar 32610, Perak, Malaysia; 3Research & Development Group, Hitachi Ltd., Tokyo 185-8601, Japan; masashi.kiguchi.py@hitachi.com; 4School of Mathematical and Computer Sciences, Heriot-Watt University Malaysia, Putrajaya 62200, Wilayah Persekutuan, Malaysia; s.dass@hw.ac.uk; 5Scientific and Industrial Research Institute of Malaysia (SIRIM Bhd.), Shah Alam 40700, Selangor, Malaysia; drfadzil@sirim.my

**Keywords:** working memory performance, workload stress, affective states, functional near infrared spectroscopy (fNIRS), haemodynamic activity, prefrontal cortex (PFC)

## Abstract

The effect of stress on task performance is complex, too much or too little stress negatively affects performance and there exists an optimal level of stress to drive optimal performance. Task difficulty and external affective factors are distinct stressors that impact cognitive performance. Neuroimaging studies showed that mood affects working memory performance and the correlates are changes in haemodynamic activity in the prefrontal cortex (PFC). We investigate the interactive effects of affective states and working memory load (WML) on working memory task performance and haemodynamic activity using functional near-infrared spectroscopy (fNIRS) neuroimaging on the PFC of healthy participants. We seek to understand if haemodynamic responses could tell apart workload-related stress from situational stress arising from external affective distraction. We found that the haemodynamic changes towards affective stressor- and workload-related stress were more dominant in the medial and lateral PFC, respectively. Our study reveals distinct affective state-dependent modulations of haemodynamic activity with increasing WML in n-back tasks, which correlate with decreasing performance. The influence of a negative effect on performance is greater at higher WML, and haemodynamic activity showed evident changes in temporal, and both spatial and strength of activation differently with WML.

## 1. Introduction

The nature of jobs is shifting towards cognitively-demanding tasks, brought about by technological advancements of the fourth industrial revolution [1]. Understanding cognitive performance at work became increasingly important because mental stress arises when the demands of the job exceeds the worker’s capability to cope [2]. While a plethora of established assessment tools have been widely used to quantify cognitive performance, these tests are typically designed to measure aspects of psychology and cognition in isolation. Measuring performance on work-related cognitive tasks can be challenging as results are often abstract and manifest only after an extended period of productivity [3]. Insights from cognitive neuroscience [4] progressively may help to shape the design of work tasks and environments as will feedback from measurements of individual cognitive response and performance. Taking direct measurements of brain activity and electrophysiological signals are an emerging alternative with which to measure work-related cognitive performance [5] given that recent technological advancements are making brain activity acquisition systems more accessible.

Among the different instrument modalities available to measure brain activity [6], functional near-infrared spectroscopy (fNIRS) and electroencephalography (EEG) stand out because both are portable instruments that may enable brain measurements to be taken at the workplace. fNIRS measures time-dependent and task-related haemodynamic responses in surface cortical regions of the brain, a signal that is comparable to the blood-oxygen-level-dependent (BOLD) signal measured by functional magnetic resonance imaging (fMRI) [7,8]. EEG measures time-dependent electrical potential variations on the scalp surface that are thought to originate from neural cortical activity [6]. Early studies show that both fNIRS and EEG can detect emotionally relevant brain activity patterns. A seminal work [9] showed that fNIRS is sensitive towards changes in task difficulty in both real-life (flight simulator) and laboratory settings (executive functions test batteries), where increased concentration of oxygenated haemoglobin (O2Hb) and decreased concentration of deoxygenated haemoglobin (HHb) were observed as the tasks became more complex. Another study [10] combining EEG, fNIRS and skin conductance response (SCR) measures showed that these physiological signals correlated with self-reported valence and arousal levels of emotion. fNIRS showed increased O2Hb activity in right pre-frontal cortex (PFC) induced by negative emotions while EEG showed delta and theta band activity was correlated with cortical haemodynamic responsiveness to negative emotions.

We study working memory as the proxy for work performance because it interfaces with several cognitive–neural systems and is well-established as a crucial construct in many higher-order cognitive functions [11]. Working memory performance reflects the capacity to retain information in an active and quickly retrievable state in the presence of divergent thought streams [12,13]. As a buffer, working memory frees the brain from the urgency of responding to stimuli in order to undertake long-term goals and empowers the brain to pursue multiple active goals to execute complex behaviours [14]. Individuals with better working memory have improved ability to control their attention and direct it to critical tasks and thereby improved ability to multitask [15]. Working memory performance is also predictive of work performance in many real-world cognitive tasks [13] and studies showed that both accuracy and response time are affected by task difficulty [16,17,18,19].

Nevertheless, relating working memory ability to work performance directly could be tricky as the interdependence between working memory and emotional states is complex [20]. Prior study showed evidence of tightly-coupled and task-selective effects of emotion on working memory [21]. Spatial working memory was enhanced by negative moods but impaired by positive emotions, whereas verbal working memory showed the opposite pattern [22]. Working memory is also believed to be involved in emotional regulation [23]. Cognitive workload that changes with mental effort is one dimension of task difficulty [24], where increased cognitive workload is expected to impair cognitive performance in a manner dependent on individual ability. Compelling theory on how task-related and situational factors interact to impact cognitive performance is lacking [4]. We see situational factors as the determinant of affect—an arousal (intensity) versus valence (direction) emotional state that impact cognition through changes in motivation [25,26,27] or the availability of cognitive resources [28]. The effect of affective states or mood (less intense but longer-lasting affect) on performance may not be uniform across different workloads and, likewise, a heavy workload may also influence mood states. Mild changes in mood affects cognition in non-intuitive ways; for example, positive moods have been shown to impair executive function [28]. These findings suggest the importance of considering the psychological aspect in workload–performance study, particularly in the high-risk industries as employees might bring their emotional sides with them to work without being aware of the consequences [29].

Since emotions play an integral part in cognition and are thought to regulate cognitive processing [30], differences in affect can significantly alter cognitive performance. Emotion can shift attention and enhance some cognitive processes like vigilance but disrupt others [20]. Emotional states are believed to alter cognition through disruption of working memory and cognitive control in favour of thoughts and actions that are congruent with mood. Negative emotional states reduce an individual’s sensitivity to reward, which biases learning and goal-directed behaviours. The effects of emotions may also endure beyond the persistence of emotional cues [18]. A recent meta-analysis of 165 studies (*N* = 7433) [4] reported that behavioural working memory performance in healthy adults was only marginally influenced by low-intensity affect but neural recruitment showed significant changes in activation of salience, fronto-parietal control networks and the temporal–occipital lobe (including the fusiform gyrus). Among individuals with poor mental health though, behavioural working memory performance was significantly affected by moods. Working memory accuracy and response time are significantly impaired by higher-intensity psychosocial stressors at high workload in n-back tasks [31].

Here, we attempt to disentangle the interrelating effects of situational and task-related factors on cognitive performance and to establish the plausibility of using direct recordings of brain activity to observe these effects. The study in [9] showed that fNIRS is sensitive towards the variation in task difficulty for both real-life work and cognitive tasks in the laboratory setting, but the relationship of OxyHb activation with task performances was not clearly observed. We anticipate that the effects of affective states on cognitive performance are reliant on the workload difficulty, and that our proposed fNIRS analyses could help to uncover the respective neural haemodynamic correlates. Our study contributes towards understanding on how cognitive stressors like task difficulty modulate the effects of situational affective stressors on working memory performance. We showed that distinct spatial and temporal haemodynamic trends characterise working memory performance under situations of increasing workload with a neutral to negative affect. These findings could help to identify haemodynamic correlations to changes in workload and affect for the implementation of real-time mental state monitoring using wearable devices in a more localised region.

## 2. Materials and Methods

### 2.1. Participants

Thirty-one (31) healthy male university students (mean age ± std. dev.: 20.74 ± 0.75 years) voluntarily participated in this study. Participants were recruited on a voluntary sampling basis and had different ethnicities (Appendix A). All participants are right-handed, and none have reported family history related to neurological or psychiatric illnesses. Aside from controlling for common confounding factors such as age [32], gender [31,33,34], handedness [35], and health status [36], we excluded participants who smoked habitually, because tobacco can affect cognitive functions in young adults [37]. Participants were randomly assigned to one of two independent groups, such that 16 individuals were allocated to the negative affective group (experimental group, EG) and 15 were assigned to the neutral affective group (control group, CG). The study took place in a sound-attenuated and air-conditioned room with minimal distraction, where participants sat on a height-adjustable chair with computer set-up resembling a workplace setting. Before the experiment, participants were given the opportunity to practice the task without viewing any mood-induced images. Participants were also given the flexibility to adjust themselves (e.g., position of the display unit, keyboard, height of the chair etc.) for maximal comfort, without interfering with the data acquisition system.

### 2.2. Visual Affective Stimuli

The International Affective Picture System (IAPS) [38] was used to induce negative and neutral affective states in EG and CG, respectively. IAPS images presented in CG include common objects in daily life, such as clothing, furniture, and accessories (mean valence: 4.64, SD: 0.32; mean arousal: 1.59, SD: 0.06); while images presented to those in the experimental group have lower valance and higher arousal ratings (mean valence: 2.12, SD: 0.45; mean arousal: 6.26, SD: 0.83), including dead animals, accident victims, and scenes of warfare etc. Images involving infants, extreme violence, or nudity were excluded due to cultural sensitivities of the participants. We referred to the normative ratings of male college students in IAPS manual while selecting the images. To ensure the efficacy of the affective contents, IAPS images selected for EG participants had a significantly lower valence (unpleasantness) and higher arousal compared with those in the CG (*t*-test, *p* < 0.001). We repeatedly displayed the IAPS images before the start of each n-back task to improve the efficacy of visual affective induction, because the study in [39] found that n-back task performance, stress-hormonal changes [31], heart rate and subjective stress index due to induced stress are modulated by time. It is also believed that the presentation of a series of affective pictures of similar valence produces emotional reactions that are either maintained or sensitised with time, but do not habituate [40].

### 2.3. WM Task Paradigm

N-back tasks with increasing WML (*N* = 0, 1, 2) were designed using e-Prime 2.0 (Psychology Software Tools Inc., Pittsburgh, PA, USA). IAPS images were shown for 10 s followed by a prompt to alert the participants at 5 s before the onset of each task. We adopted block design as it has a higher statistical power to detect task-related haemodynamic activity and more robust to individual uncertainties in temporal haemodynamic response functions as compared to the event-related paradigm [41]. In each WML condition, participants completed three blocks of 40 s of task interspaced by a 15 s rest interval as illustrated in Appendix A. This rest duration allows sufficient time for haemodynamic activity responses to recover from sustained elevation due to the effortful task [42].

During each task, 20 characters (i.e., A, B, C, D, E, and X) were presented in pseudorandom order for 40 s (2 s window per character, fading slowly after 1 s to alert the participants). Participants were required to respond as quickly as possible using the numerical key on a standard-size keyboard before the next character is shown. In the case of the 0-BT, participants simply identify the target (character ‘X’) by pressing “1” and reject other characters as non-targets (by pressing “2”). For higher WML, participants are required to decide if the current character matches the previous *N*th character. We measured task performance accuracy and response time as behavioural markers. Statistical analysis using a two-way ANOVA mixed model with a between-subject factor (affective states) and a within-subject factor (WML) was performed using the Statistical Package for the Social Sciences (SPSS version 23, IBM Corp., Armonk, NY, USA).

### 2.4. Haemodynamic Measurement

We used the OT-R40 Optical Topography system (Hitachi Medical Corporation, Tokyo, Japan) to measure haemodynamic responses in PFC. The system is identical to the commercial ETG-4000 series based on continuous wave dual-wavelength near-infrared spectroscopy (NIRS: 695 nm and 830 nm), at a sampling rate of 10 Hz. A total of 15 sources and 15 detectors were arranged in a 3 × 10 layout to cover the PFC region (Appendix A). Each source-detector distance is kept at 30 mm, which lies within the recommended range of 16–32 mm for observing working memory task-related haemodynamic activity on the adult’s forehead [43]. The average power of each source was 2 mW (for each wavelength), and the dual-wavelength lights were irradiated on the measuring site through an incident optical fibre bundle [44]. During the placement of fNIRS probes, we aligned channel 43 (lowest measuring channel at mid-PFC) on the Fpz location accordance to the 10–20 system. To estimate our measuring channels’ mapping with the Broadman area, we referred to an adult fNIRS study that utilised a similar system on the PFC region [45], in which the channel’s locations were probabilistically estimated and anatomically labelled using the standard brain space. Subsequently, we defined the region of interests as shown in Appendix A, where the shaded channels are closely associated to bilateral medial prefrontal cortex. We then grouped channels 9, 18–19, 27–28, 36–38 and 45–17 from the left and channels 1, 10–11, 20–21, 29–31 and 39–41 from the right as the lateral PFC region (LRPFC). Non-shaded channels were associated to the fronto-polar and dorsolateral prefrontal cortical regions (FP).

### 2.5. fNIRS Data Analysis

As early study has shown that the light picked up by the fNIRS detector is correlated to haemoglobin concentration (Hb) and scattering depth in the brain [46], we used the Platform for Optical Topography Analysis Tools [47] to convert the channel-wise NIRS detected light intensity into changes of oxygenated and deoxygenated haemoglobin concentrations (OxyHb and deOxyHb signals) based on the modified Beer–Lambert law [48]. These converted signals are expressed as the product of haemoglobin concentration change (mM) and optical path length (light scattering path in each channel, mm) in the unit of mM.mm. We started pre-processing the raw Hb signals by identifying the motion artefacts, defined by sudden change of Hb signal amplitude larger than 0.4 mM.mm over 2 successive samples (200 mS) [49]. Next, we used band-pass filtering (0.008~0.12 Hz, 5th order Butterworth) to remove DC drift and reduce non-brain related physiological components such as heart rate in the signals [43]. The higher frequency limit is larger than two-fold [7] the duration for n-back task block (1/52 s = 0.019 Hz). For each WML condition, 3 blocks of data were recorded. We averaged the block signals in each channel with 5 s pre-scan, 10 s IAPS image presentation, 5 s pre-task, 40 s task, and a 10 s post-task period as shown in Appendix A. The blocks which were contaminated by motion artefacts were omitted from the analysis. Next, we applied baseline correction to each channel using the linear fitting method based on the data points in the pre-scan and post-task periods. A moving average filter of 30 datums (3 s) was applied to smooth the signals by decreasing high frequency noise.

We paid more attention on OxyHb signals, because it is more directly associated with brain activity [50], and the dominant effect of the n-back task has been reported [36]. To examine if PFC sub-regions contribute differently towards the IAPS images and n-back tasks, we performed paired a *t*-test on each channel to compare the OxyHb magnitudes of (i) IAPS versus Pre-scan and (ii) n-back tasks versus Pre-task, respectively, under merged WML conditions. We indicate the t-values and significant activated and deactivated channels (2-tail, Bonferroni-corrected α < 0.05/47) during IAPS image presentation and while performing n-back tasks on the PFC topography layout, for both affective groups. The OxyHb changes during the IAPS image presentation period and n-back task blocks were analysed independently in order to consider brain activation towards cognitive and emotional processes differently [34].

To obtain a sensible task haemodynamic response across participants, we performed a baseline correction for both OxyHb and deOxyHb temporal signals with respect to the task activation block as shown in Appendix A. The 2 s pre-task and the last 2 s of the post-task period were used for baseline correction using the linear fitting technique in each channel, to eliminate subsequent stacking effects of haemodynamic activation while viewing the affective images [51]. We defined the task activation period as a 25 s window (15 s after task onset to the end of the task), after taking the delay of haemodynamic response [49] and our task duration into consideration. A related study also found that fNIRS signals with a 25 s window can be used to robustly quantify different levels of workload for n-back tasks [52]. We analysed the channel-wise mean OxyHb and deOxyHb concentration changes during n-back tasks in PFC with respect to affective states and WML conditions on the optical topography map. The signal means are calculated as follows:(1)$$HbX_{mean}\left(i,j\right)=\frac{1}{N_{w}}\overset{t_{2}}{\underset{t=t_{1}}{\sum }}\Delta HbX\left(t\right)$$
where the subscript w denotes task activation window; *t*_1_ and *t*_2_ denote the start and end time of the window, respectively. *HbX* refers to either OxyHb or deOxyHb data. The iteration is repeated for *i*th subject and *j*th channel.

To distinguish if the changes of mean haemodynamic concentration with affective state and WML are due to variation of activation area or strength (intensity), we derived features to consider haemodynamic changes by area and strength. First, we identify the n-back tasks’ channel-of-interests in PFC for each participant by comparing the OxyHb magnitude during task activation window versus pre-task using a *t*-test (2-tail). Bonferroni corrections were applied to 47 channels (α < 0.05/47) to exclude false positive error. We then compute the intensity of task activation for each individual (*i*) as follows:(2)$$OxyHb_{intensity}\left(i\right)=\frac{\sum_{j=1}^{N_{Act}}OxyHb_{mean}}{N_{Act}}\;$$
where $N_{Act}$ is the total number of activated channels for subject *i*th. In any case, where *j* = 0 (i.e., no activated channel within the specific area of interest) the computation is omitted. Next, we estimated the total positive area under the curve (AUC) for the subject-averaged OxyHb temporal signal at LRPFC using the trapezoidal function as follows:(3)$$AUC_{LRPFC}\left(i\right)=\{\begin{array}{c} \;\;\;\;\int_{a}^{b}f\left(t\right)\;dt=NaN,\;\;\;\;\;\;\;\;\;\;\;\;\;\;\;\;\;\;\;\;\;\;\;\;\;\;\;\;\;\;\;\;\;\;\;\;\;\;\;\;\;\;\;\;\;\;\;\;\;\;\;\;\;\;\;\;\;\;if\;f\left(t\right)\leq 0\\ .\\ .\\ \int_{a}^{b}f\left(t\right)dt|≅\frac{1}{2}\overset{N}{\underset{n=1}{\sum }}\left(t_{n+1}-t_{n}\right)\left[f\left(t_{n}\right)+f\left(t_{n+1}\right)\right],\;\;\;\;\;otherwise \end{array}$$
where *f*(*t*) is the averaged OxyHb temporal signal among significantly activated channels in LRPFC with respect to subject *i*th, *a* =$\;t_{2}$ < $t_{2}$ <  ...  < $t_{N}$ < $t_{N+1}$ = *b*, and ($t_{n+1}-t_{n}$) is the spacing between each consecutive pair of points. We only considered the total AUC whenever the OxyHb signal crossed above baseline (OxyHb > 0) from time a to b. Lastly, we performed statistical comparison for the selected spatial and temporal haemodynamic features in 3 WMLs × 2 affective groups as follows: Area of activation (number of significant activated channels) in PFC, FP and LRPFC;Strength of activation (OxyHb intensity) in PFC, FP and LRPFC;Time taken from n-back task onset to OxyHb’s highest peak at LRPFC;Total OxyHb positive area under curve at LRPFC.

We could not apply the ANOVA model for testing the statistical significance of fNIRS data because of the imbalance in the number of activated channels between participants under different profiles of workload and affective state (Appendix A) which compromises statistical power. Instead, we performed a combination of pairwise statistical comparisons followed by corrections for multiple comparisons. Statistical pairwise comparison for (i) and (ii) were assessed by independent *t*-tests (2-tail). In any case when the homogeneity of variance was not assumed, a Welch *t*-test was applied. The temporal data in (iii) and (iv) showed that the equality of variance assumption was met in Levene’s test (sig. > 0.05). However, there appeared to be several outliers (>1.5 interquartile range) in the 0-back condition (<5 cases in each group for AUC) that we believed were genuine observation (upon visual inspection of individual OxyHb signal at LRPFC during no-load condition), and hence were retained for analysis. In addition, these data (0-back) in (iv) did not meet the assumption of normality for parametric comparison, as assessed by the Shapiro–Wilk test (sig. < 0.05). We opted for the Mann–Whitney U test to determine if the temporal time-to-peak and total positive area under the curve (AUC) features were significantly different between neutral and negative affective states. Bonferroni corrections applied for multiple comparisons to prevent type I error, unless otherwise stated.

## 3. Results

### 3.1. Negative Affective State Undermine Working Memory Performance and the Effect Is More Pronounced as Task Difficulty Increases

Two-way mixed ANOVA with Greenhouse–Geisser correction [53] revealed a significant interacting effect between WML and the affective state on accuracy (*F*_(1.425, 41.337)_ = 3.827, *p* < 0.05, partial *η*^2^ = 0.117), but not response time (*F*_(1.355, 39.303)_ = 1.829, *p* = 0.183, partial *η*^2^ = 0.059). Follow up analysis showed that the simple main effect of WML on accuracy is highly significant in both the neutral affective state (*F*_(1.465, 20.515)_ = 42.30, *p* < 0.0001, partial *η*^2^ = 0.751) and negative affective state (*F*_(1.352, 20.28)_ = 50.624, *p* < 0.0001, partial *η*^2^ = 0.771). On the other hand, response time changes significantly with WML (main effect: *F*_(1.355, 39.303)_ = 41.370, *p* < 0.0001, partial *η*^2^ = 0.588), but not the affective state [*F*_(1,29)_ = 1.824, *p* = 0.187, partial *η*^2^ = 0.059]. We referred to [54] for benchmarks to define small (*η*^2^ = 0.01), medium (*η*^2^ = 0.06), and large (*η*^2^ = 0.14) effects.

Participants had comparable task accuracy during no load conditions (0-back), regardless of affective states. Under the negative affective state, participants underperformed in accuracy with increasing workloads, whereby the greatest disparity in performance occurred at the highest workload condition as shown in Figure 1. Post hoc analysis using pairwise comparison revealed that the negative affective group performed significantly poorer than the neutral group during the 2-back task (*F*_(1, 29)_ = 4.242, *p* < 0.05, partial *η*^2^ = 0.128). Response times in the negative affective state are consistently shorter in duration than in the neutral state, but not significantly different. The pairwise comparisons also revealed that the response time in the 2-back task was not statistically increased from the 1-back task (*p* = 0.062) in the negative affective group, unlike the neutral control group that showed significant changes (*p* < 0.05) between all WML conditions.

### 3.2. Distinct Trends of Spatial and Temporal Haemodynamic Activity Are Correlated with Working Memory Performance with Changes in Affective State and Working Memory Load

Channel-wise analysis in Figure 2a showed that PFC sub-regions contributed differently while participants were subject to different procedures (see Appendix A). The active sub-regions corresponding to viewing IAPS images and performing n-back tasks were more established at FP and LRPFC, respectively. Significant activated channels corresponding to n-back tasks were also likely to be significantly deactivated during IAPS, and vice versa. While viewing the IAPS images, participants under the negative affective state had fewer activated channels as compared to participants under the neutral state, particularly at right FP. The detailed comparisons are available in Appendix A.

We find distinct spatial and temporal patterns of n-back tasks’ haemodynamic activity in channels-of-interest (Appendix A) that correlate with different profiles of affective states and workloads. In the neutral state, mean task activation increases with WML in LRPFC, whereas in the negative affective state, task activation decreases with increasing WML as shown in Figure 2b. On the other hand, haemodynamic activity in FP region appears to change with affective states, not variation in WML. We also find that LRPFC activation is more asymmetric on the left PFC during n-back tasks, particularly among the negative affect group.

Statistical comparisons in Figure 3 show that the haemodynamic area and intensity of activation in whole-PFC is not significantly different between affective groups, once Bonferroni–Holm corrections were applied to correct the family-wise error rate across WML conditions. Region-wise analysis revealed that the intensity of activation in both FP and LRPFC increase monotonically as task difficulty increases under the neutral state. As workload increases, FP first shows increased area of activation and subsequently a reduction in neural recruitment. In the negative affective state, we find a reciprocal pattern where intensity of activation in both FP and LRPFC monotonically decreases with increasing task difficulty. The largest disparities in area and intensity of activation between affective groups occurred at FP during medium the (1-back) and highest (2-back) WML, respectively (*p* < 0.05, large effect: Cohen’s d > 0.8. Details in Appendix A).

Although the haemodynamic activation in LRPFC was not statistically significant (Figure 3), the disparity between affective groups is deemed evident during the 2-back task, indicating a moderate effect in changes of area (*p* = 0.079, d = 0.654) and intensity of activation (*p* = 0.071, d = 0.583). Temporal analyses on LRPFC’s OxyHb signals (Figure 4) further revealed that under no-load conditions (0-back task), the neutral and negative groups had comparable OxyHb time to peak, whereas under load conditions (1-back and 2-back), the signals’ time to peak between the neutral and negative affects was strikingly different (*p* < 0.05, large effect: Cohen’s d > 0.8) as shown in Figure 5. In negative affective state, the total positive AUC decreases monotonically with increasing WML and showed the greatest disparity versus neutral state during the 2-back task (medium effect, Cohen’s d = 0.68). We referred to [54] for benchmarks to define small (d = 0.2), medium (d = 0.5), and large (d = 0.8) effects. The detailed statistical result is attached in Appendix A.

fNIRS data from channels-of-interest (Appendix A) showed that the temporal changes of OxyHb and deOxyHb in bilateral PFC are negatively correlated during the n-back tasks activation window (Figure 4). This haemodynamic pattern agrees with the established fNIRS signal processing guide [55] which indicates neurovascular coupling [56] evoked by a working memory task. Sustained activation within the task period also reflects the general operational processes associated with task difficulty and mental effort, aside from working memory [42]. At the individual level, we found that the post-task OxyHb signals have a tendency to reach a below baseline level in the LRPFC region indicating post-stimulus undershoot effects after an effortful task [41]. Our protocol design allows sufficient time (>10 s after each task block) for haemodynamic activity to return to baseline during the resting state.

## 4. Discussion

We found interesting distinctions in the mechanism of neural activation at different brain regions during the working memory task as workload increased. In the LRPFC, increased mental effort correlated with greater intensity of activation whereas, in the FP-PFC, area of activation is dominantly modulated together with minor elevation of intensity. In addition, the pattern of activation in FP-PFC is not monotonic with workload but resembles an inverted-U [58,59,60,61]. Increasing task difficulty appears to modulate neural recruitment, initially by increasing resources at moderate task loads but subsequently reducing neural recruitment at heavy workloads. Workload can act as a stressor at high loads [5], presumably when cognitive capacity is no longer sufficient to adequately execute the task [13]. Our data suggest this possibility because the rate of decline in both measures of working memory performance (poorer accuracy and longer response time) is more evident with elevated workload. We anticipate that workload-related stress will compromise performance of goal-directed information retrieval more significantly than information persistence, because FP-PFC recruitment is depressed.

The inverted-U concept postulates that mild to moderate stress improves performance and we see evidence of group-wise improvement on the 0-back and 1-back task during the negative affective state (comparable accuracy but response time is improved). Severe stress is also expected to degrade performance precipitously, and we see rapid decline in accuracy with increasing workload under the negative affective state. The largest disparity in performance between negative versus neutral affective state was observed during the most difficult task (2-back). This trend suggests that the negative affective state acts as a stressor on cognition, but only as an interacting effect with WML. Our results agree with an early study which suggests that the non-neutral mood state imposes a psychological load over the cognitive resources and will likely impair performance because of finite mental capacity [28]. The effect of compounding affective and workload stressors is much more detrimental to performance than either stressor acting alone. Among mentally ill subjects, working memory performance was also more significantly degraded by negative mood [4] as compared to healthy adults. For example, schizophrenic patients had significantly poorer performance and reduced DLPFC activity in the highest load task (3-back) [62]. Taken together, this evidence suggests that cognition may be significantly impaired by the negative affective state only when acting in concert with other stressors like higher-workload or illness. It is likely that the effects will also be modulated by other factors such as task alignment of the emotional stimuli, age and individual differences in emotional regulation [4]. New studies are needed to explore the spectrum of factors and their interactions because there still is a lack of neural activity data in personality-driven variations and in a variety of mental illnesses.

Irrespective of performance, emotional factors consistently modulated neural activation during a variety of working memory tasks. Meta-analysis of 33 fMRI working memory studies (*n* = 683) in healthy adults found reduced activation of right DLPFC and increased activation of left VLPFC and OFC for working memory tasks under negative as compared to neutral moods [4]. The study argued this as an evidence for competitive allocation of neural resources away from executive control to perceptual processing tasks [20,25,63,64] which predicts enhanced activation of salience networks (includes VLPFC and amygdala) and reduced activity of the frontoparietal control network (includes DLPFC). Increased activation of default mode network in tandem with reduced DLPFC activation was also observed under negative mood-stressed subjects [19]. Neural processing of emotion and cognitive processing are tightly integrated [21] and it has been argued that distinct but closely meshed neural circuits in the frontal cortex (MPFC) are separately responsible for emotional regulation and cognitive processing [65].

Consistent with fMRI studies, we find reduced DLPFC activation intensity during the negative state. Prior fNIRS studies of verbal working memory have also found reduced DLPFC activation under negative moods [66,67] but it was not replicated for spatial working memory tasks. Our contribution is to demonstrate that haemodynamic intensity reduction occurs in a workload-dependent manner. There is only a slight reduction in DLPFC area of activation. This effect is striking because previous fNIRS studies have shown that viewing images per se to induce strong negative emotions (without working memory task) also increased OxyHb activation in VLPFC [51]. During the negative state, FP-PFC activation area and intensity are depressed and no longer modulated by workload. This result possibly indicates reduced neural recruitment and mental effort for goal-directed information retrieval because negative mood causes neural resources to be redirected from the working memory task. According to the theory of competitive reallocation, we might expect concomitant increase in neural haemodynamic activity in frontal cortex regions associated with emotional regulation. However, we find broad deactivation instead which suggests that the negative affective state compounded with difficult workloads inhibits neural activation in PFC regions associated with emotional regulation. It is also possible that depressed activation is linked to loss of motivation (withdrawal) under difficult workload and emotional situations.

In an fMRI study [42], greater DLPFC activation was observed as working memory load increased (more items to recall or longer memory retention period). Individuals with poorer response time showed increased DLPFC activation with reduced-workload dependent increase [68]. These findings suggest that DLPFC activation is linked to cognitive effort. Higher activation is consistent with greater effort [42], whether it be to compensate for increased task difficulty or poorer cognitive efficiency. fNIRS studies also show correlations between increased OxyHb activation and decreased HHb activity with increased task difficulty [9,52,69,70] and have proposed fNIRS as a useful tool to measure mental effort. Furthermore, an earlier fNIRS study [71] reported gender-specific haemodynamic activation in PFC—that males showed bilateral activation with slightly left-side dominance, whereas females showed largely left activation. Our fNIRS finding on male participants shows a similar trend in bilateral activation (Figure 2b) and group analysis of neural haemodynamic activity under neutral affective state shows both increasing intensity and area of activation (Figure 3), as the n-back task becomes progressively more challenging.

Increased DLPFC and MPFC activation was observed in fMRI studies on verbal working memory tasks using emotion words [72] and in another working memory task requiring visual identification of emotion [73]. Unlike our study where pictures where only used as an external distractor to induce affective states, the emotion words and pictures in these studies induced mood and provided additional contextual information for memorisation. We argue that emotional content was aligned to the working memory task and did not detract from the cognitive effort of working memory. Therefore, we are unsurprised by the different patterns of neural activation. Increased DLPFC activation suggests additional mental effort which may be linked to increased arousal or motivation due to task-aligned emotional content. A related fNIRS study (*N* = 20) also found increased OxyHb activity in the VMPFC region with increasing workloads under the negative affective state [34], but the difference in task performance is not evident. We believe that the disparity in findings may be due to several methodological differences. First, this study rapidly alternates between neutral and negative affective states among participants within a single experiment session. We think that it is difficult to control or maintain the participants in a consistent emotional state under such settings, as mood itself is not a simple dichotomous phenomenon [28]. Switching between the n-back task with different WML demands may also incur additional mental resources and increase the error rate among participants [74]. Moreover, the temporal gap between IAPS presentation and the n-back task was too brief (1.3 s instruction cue) where the haemodynamic responses due to affective images [51] may not have sufficient time to return to baseline. Under these conditions, it would be difficult to ensure task-related haemodynamic activation was separated from emotional processing. We postulate that swift WML switching might also act as an additional stressor to participants.

Temporal analyses at LRPFC revealed that the signals’ time to peak and AUC are markedly different between affective groups, although changes in activation area and intensity at LRPFC were not statistically different. Interestingly, the times to peak between affective groups were highly comparable during 0-back task (*p* > 0.99, d = 0.04), but strikingly different (*p* < 0.05, d > 0.8) during the 1-back and 2-back tasks. Since the OxyHb signals’ time to peak feature was relatively uncommon in fNIRS studies [75], our finding suggests that it could be a useful marker for affective states when moderate-to-high working memory loads were engaged. On the other hand, larger OxyHb AUC may indicate a greater vigilance as it was significantly correlated with the amount of physical exercise [76] and restful sleep [77]. fNIRS study found significant lower OxyHb AUC among sleep-deprived subjects, who had compromised cognitive performance and disturbed mood state related to vigour, as assessed by the Profile of Mood States [77]. We found that the OxyHb AUCs between affective groups are comparable during the 1-back task. However, participants under the influence of the negative affective state showed relatively higher and lower AUCs during no-load (0-back) and high-load (2-back) tasks, respectively. Under the no-load condition, the negative affect may impose additional psychological load [28] where participants are to be more alert, evidenced by slight improvement in accuracy and lower response time. However, high WML (2-back) acting in concert with negative affect and increased time on task (chronologically: 0,1,2-back) are likely to have exceeded participants’ cognitive capacity to execute the given task [13], evidenced by significantly poorer accuracy. Our findings agreed that emotion can shift attention and enhance some cognitive processes like vigilance but disrupt others [20], depending on individual capacity and external factors. Taken together, we showed that haemodynamic analyses which take the activation area, strength and temporal properties into consideration, together with different PFC sub-regions, are necessary as the interaction of affect and cognition is nuanced [30].

Brain activation patterns appear to have a gender-dependent contribution. Studies from the verbal working memory task showed that male subjects had significantly higher OxyHb and TotalHb changes in bilateral PFC, as compared to female subjects who showed predominantly left-PFC activation, even though their behavioural performances (accuracy and response times) were comparable under various workload conditions [71]. When exposed to negative visual stimuli, female subjects showed more intense OxyHb responses while male subjects had no significant change [78]. One major limitation of our exploratory study is that we recruited male-only participants, so our findings are limited to young adult males with high educational attainment. However, our participants were recruited from diverse ethnic backgrounds. We also find predominantly bilateral PFC activation in working memory studies involving first authors from China [19], Japan [66], and Germany [72] that relate to acute stress (females only), naturalistic mood states, and emotional stimuli (males only), respectively. These findings suggest that gender and ethnicity may not be the critical biases for an emotion–cognition study.

For our future work, we wish to develop a methodology for brain-guided measurements to inform the design and scheduling of work-related tasks. We believe portable brain signal acquisition systems like fNIRS and EEG have an important and untapped potential to improve the neuroergonomics of work-related tasks and, thereby, enhance worker productivity, well-being and safety. The study reported in this paper is a preliminary exploration towards our long-term goal. Future experiments should include studies over different genders and spanning the entire working age of healthy adults with different educational attainments and cognitive capacities. We also anticipate replicating our analysis on a variety of different executive and cognitive tasks besides working memory, in order to generalise our findings over the range of cognitive tasks that a worker may regularly employ in the course of a job.

## 5. Conclusions

We established that cognitive performance and haemodynamic responses towards different affective states are workload dependent, and that direct brain measurement using fNIRS helps to unravel such interaction when participants are engaged with moderate–high working memory loads. We also found that PFC sub-regions contribute differently towards affective stimuli and n-back tasks, under different profiles of workload difficulty and affective states. In FP, the negative affect leads to significant depressed neural recruitment by activation area and intensity, during 1-back and 2-back tasks, respectively (both a showed large effect, d > 0.8). Although the activation area and strength in DLPFC were not statistically different between affective states and across WMLs, temporal analyses revealed distinctive patterns in the signal’s time to peak and AUC The negative affect had significantly shortened OxyHb time to peak in LRPFC (large effect, d > 0.8 during 1-back and 2-back), and reduced OxyHb AUC during 2-back task (medium effect, d > 0.5). We postulate that these temporal features correspond to the affective marker and mental alertness, respectively. New studies are needed to correlate these temporal markers with established assessments (e.g., the positive and negative affect schedule or mini mental state examination) on a larger sample to reaffirm the causality. Overall, we showed that a holistic analyses approach that considered fNIRS data from aspects of activation area, intensity, temporal properties, and PFC sub-regions are important. Our study also suggests that fNIRS is indeed helpful to uncover the interaction of working memory loads and affective states, as behavioural performance data only indicate that the negative affect impaired task accuracy (medium effect: partial *η*^2^ = 0.128; but not response time) during the highest WML condition (2-back). We anticipate that the application of a wearable fNIRS device to monitor mental states in critical workplaces is becoming more feasible, as reliable and straightforward (without complex computational load) markers for affect and cognition are progressively established at a more localised PFC region.

## Figures and Tables

**Figure 1 brainsci-11-00935-f001:**
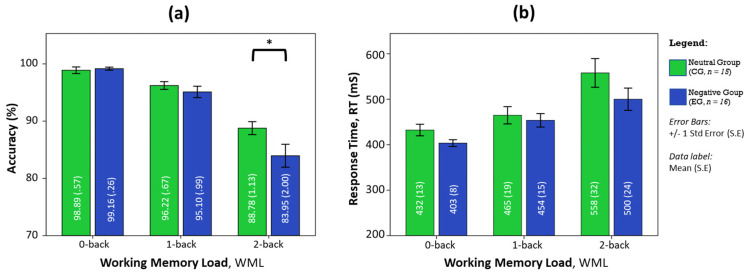
The influence of affective states on (**a**) task accuracy and (**b**) response time with WML. ANOVA post-hoc analysis of pairwise comparisons was performed using the F-test with significance at * *p* < 0.05, error bars are S.E.

**Figure 2 brainsci-11-00935-f002:**
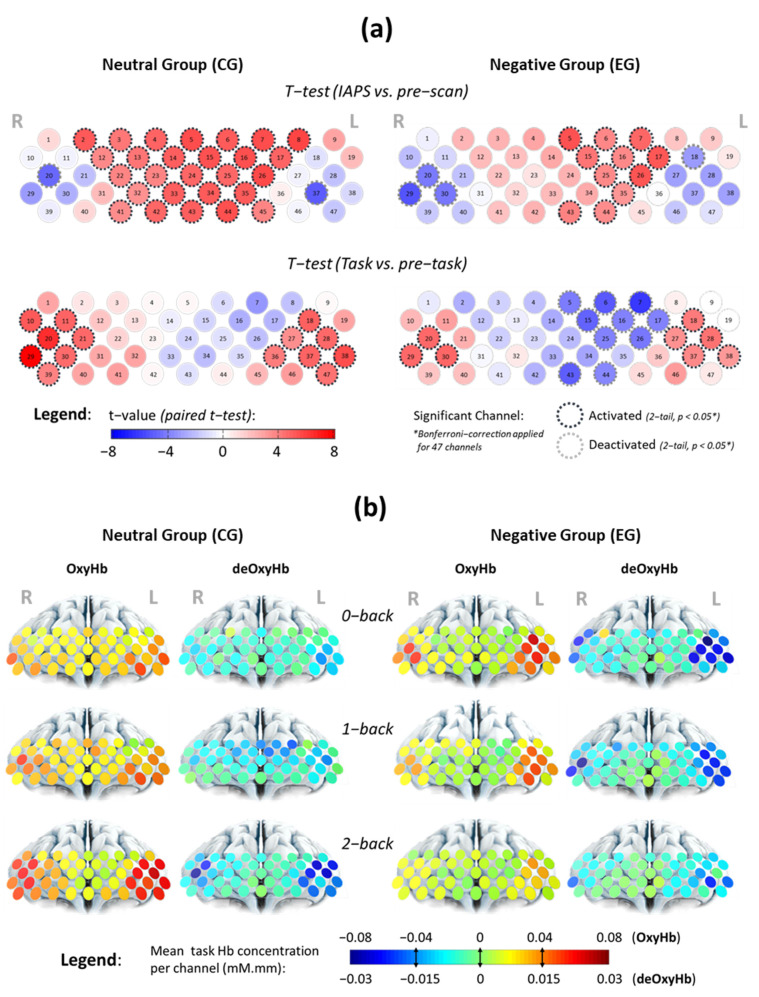
Analysis of channel-wise contrast using (**a**) a paired *t*-test under merged WML conditions to determine if the mean OxyHb changes towards IAPS images and WM tasks are region-specific and how (**b**) mean OxyHb and deOxyHb changes with affective states and WMLs during n-back tasks. Intensifying red in (**a**) indicates a stronger positive contrast of signal amplitude, while intensifying blue indicates negative changes. Pale colour indicates comparable signal level between both periods. Significant activated and deactivated channels were circled with black and grey dotted lines, respectively, after Bonferroni–Holm’s corrections. Warmer colours in (**b**) indicate higher activation for OxyHb during WM tasks, while colder colours indicate greater reduction in deOxyHb.

**Figure 3 brainsci-11-00935-f003:**
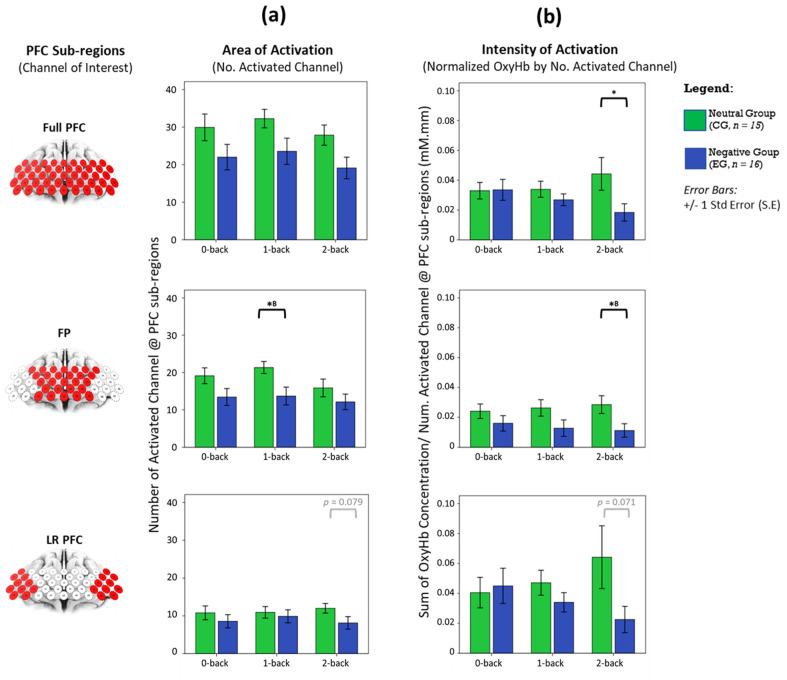
Does haeomodynamic activity vary due to changes in area of activation or strength, and are there any dominant effects neuroanatomically? We plotted (**a**) the number of significantly activated channels and (**b**) normalised activation from significantly activated channels against WML and affective states in whole PFC, FP and LRPFC regions. Statistical analysis performed between CG and EG using an independent *t*-test (2-tailed), with significant * *p* < 0.05 without family-wise error rate correction and (^B^) with Bonferroni–Holm’s correction (α = 0.0167). Error bars indicate the standard error (S.E).

**Figure 4 brainsci-11-00935-f004:**
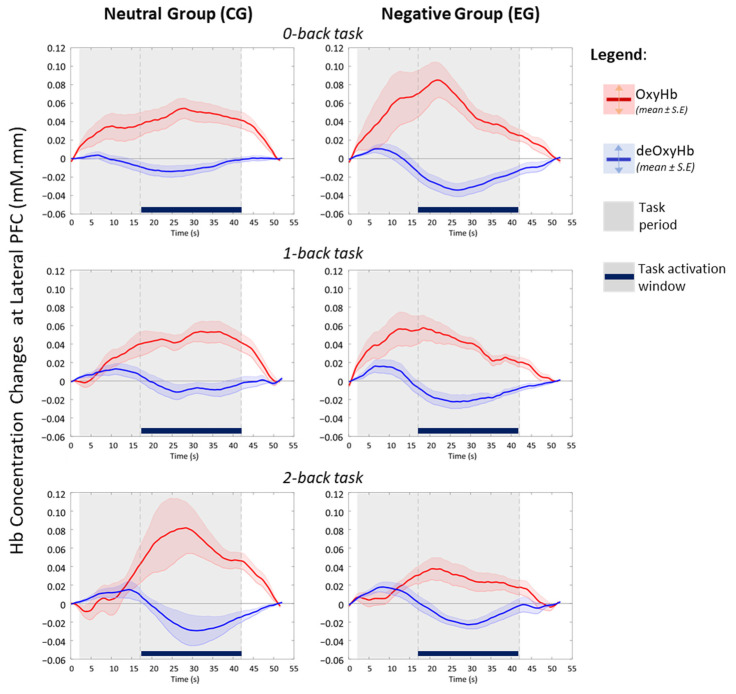
Averaged temporal haemodynamics from significant activated channels in LRPFC during n-back tasks (adapted from [57] with written permission).

**Figure 5 brainsci-11-00935-f005:**
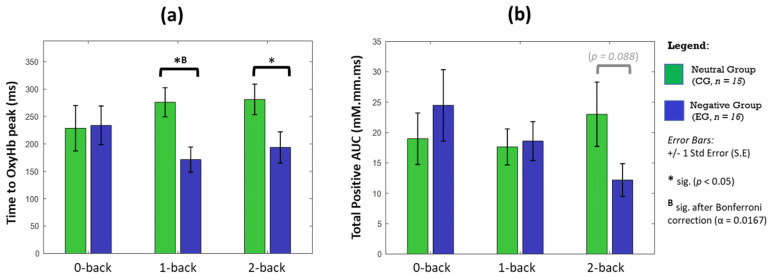
OxyHb temporal properties of significant activated channels in LRPFC during n-back tasks. We plotted the OxyHb signal’s (**a**) time taken to the highest peak from task onset, and (**b**) the total area under the curve (AUC) where OxyHb > baseline zero for each affective state and WML. Statistical analysis performed between CG and EG using an independent *U*-test (2-tailed), with significant * *p* < 0.05 without multiple-hypothesis test correction and (^B^) indicates Bonferroni–Holms correction applied (*α* < 0.0167). Error bars indicate standard error (S.E).

## Data Availability

The datasets generated and analysed during the current study are available from the corresponding author on reasonable request. The data are not publicly available due to privacy and ethical concerns.

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
