# Peer review of "Working Memory Performance under a Negative Affect Is More Susceptible to Higher Cognitive Workloads with Different Neural Haemodynamic Correlates"

_brainsci, 2021, doi:10.3390/brainsci11070935_

Round 1

Reviewer 1 Report

Undoubtedly, the effect of stress on task performance is complex, too much or too little negatively affects performance; and there exists an optimal level of stress to drive optimal performance. Task difficulty and external affective factors are distinct stressors that impact cognitive performance. 

My comments to the article are as follows:

- I propose to extend the background to the article by referring to the methods of analysis and acquisition of signals flowing from the human brain in the Introduction. For example, you can refer to: Methods of Acquisition, Archiving and Biomedical Data Analysis of Brain Functioning, Biomedical Engineering And Neuroscience, Book Series: Advances in Intelligent Systems and Computing, Springer from 2018.

In addition, in the field of research on stress, I propose to refer to the influence of music on stress, verified by signals from the brain. For example, I propose to quote: The Impact of Different Sounds on Stress Level in the Context of EEG, Cardiac Measures and Subjective Stress Level: A Pilot Study, BRAIN SCIENCES from 2020.

- Please provide arguments regarding the choice of such and not another statistical method - ANOVA.

- Please provide information on what basis did you choose the students for the research? How did you approximate their population?

- I propose to extract part of the Results section in Discussions. In my opinion, this is missing from the article.

- As part of Conclusions, I propose to expand the description of plans for the future of the conducted research.

Reviewer 2 Report

The manuscript titled “Working Memory Performance under Negative Affect is More Susceptible to Higher Cognitive Workloads with Different Neural Haemodynamic Correlates” proposed an investigation on the neural haemodynamic correlates in the prefrontal cortex (PFC) of the interaction between negative affect and working memory load (WML) in a working memory performance task, using functional near-infrared spectroscopy (fNIRS) neuroimaging technique. Thirty-one healthy male university students were randomly assigned to a negative affect group or to a neutral affective group, watching pictures from International Affective Picture System, and completed three n-back tasks (0, 1, 2 n-back condition). Accuracy, time, spatial and temporal haemodynamic features on participant’s forehead were recorded. Results showed that there were more dominant haemodynamic changes towards affective stressor and workload related stress in the medial and lateral PFC, respectively. Moreover, distinct affective state-dependent modulations of haemodynamic activity were found with the increasing of WML in n-back tasks, correlating with decreasing performance. Authors concluded that negative affect influences performance at higher WML, and haemodynamic activity showed evident changes in temporal, and both spatial and strength of activation differently with WML. Authors discussed their results in light of previous literature and gave hints for future research.

I carefully read the manuscript, and I think it may be of interest for the readers of Brain Sciences. The manuscript is very well-written and properly addresses the interesting issue of the relationship between PFC activation and cognitive workload mediated by affective state. It is also relevant that Authors used a non-invasive and mobile neuroimaging technique, which allows to detect changes in brain activation during the performance of various everyday tasks.

It was a pleasure for me to read the manuscript. The introduction section as well as the aims of the study are clear and detailed. The methodology is very rigorous and accurate, as well as the explanations provided in the discussion section.

I have only two questions/considerations:

1) Why did you choose to select only male participants?

2) It would be useful for the reader to insert this choice as a limitation of the present study, and it would be interesting to know whether the other relevant studies you cited in the discussion section found a gender effect (for example, Fishburn et al. (2014) reported that “fNIRS has demonstrated sensitivity to group differences in activation during working memory, based on gender (Li et al., 2010)”), and whether you results show similar patterns with those obtained in male groups.

Author Response

We thank reviewer 2 for the thoughtful comments.

We were only able to recruit and sustain interest from male-only paid volunteers in our university. The choice for male-only participants was not a deliberate one and is now highlighted as a limitation of our study in the Methods Section 2.1 and in the final paragraph of the Discussion. We have also added Supplementary Table S1 to summarize the demographic features of the volunteer population. Unlike in Western university campuses, there are cultural and religious barriers to female volunteers in Malaysia. The majority of our women students wear the Muslim head covering and would not uncover their head for the fNIRS experiment because of the presence of the male first author, who conducted the data collection activity.

We have included a new paragraph just before the end of the Discussion section, which references literature reporting differences in the laterality and intensity of activation between male and female subjects. Our results for male-only subjects are concordant with (Li et.al, 2010) showing bilateral PFC activation in Figure 2b.

Finally we wish to express our appreciation to Reviewer 2 for the positive and encouraging comments. It is not often we encounter such generous and affirmative comments. We would like to express our deeply felt gratitude and to just say, how much this means personally to us - thank you.  

Round 2

Reviewer 1 Report

Dear Authors, 

Thank you for the changes made. Thank you for the replies sent too.

Currently, I have no comments on the article.